materials science

alkali−carbonate reaction, dolomitic aggregates, expansion, microstructure

**Author for correspondence:**
Min Deng
e-mail: dengmin@njtech.edu.cn

This article has been edited by the Royal Society of Chemistry, including the commissioning, peer review process and editorial aspects up to the point of acceptance.

# Effects of curing condition and particle size of aggregate on the expansion and microstructure of dolomitic aggregates cured in TMAH solution

Bin Yang, Xiaoxiao Chen and Min Deng

College of Materials Science and Engineering, Nanjing Tech University, Nanjing, People's Republic of China

BY, 0000-0002-1407-4794; MD, 0000-0002-3894-3958

In this paper, the modified microbars prepared by dolomitic aggregates with three kinds of particle size and self-made cement without $K^+$ and $Na^+$ were cured in 1 and 2 N tetramethyl ammonium hydroxide (TMAH) solution at 20°C, 60°C and 80°C, respectively. TMAH was used as curing solution to exclude the expansion contribution of alkali−silica reaction. Effects of the concentration of TMAH solution, curing temperature and aggregate grain size on the expansion of dolomitic aggregates were systematically investigated to determine the expansion characteristics only caused by alkali−carbonate reaction (ACR). Expansion of modified microbars cured in TMAH solution was measured. The porosities of original and reacted aggregates were also measured. Microstructural studies were carried out by scanning electron microscopy (SEM) and thermo-gravimetric (TG) analysis. The results showed that the aggregate grain size and curing temperature can influence the expansion of modified microbars significantly. When the modified microbars prepared by aggregates with 2.5−5 mm grain size and cured in 1 N TMAH solution at 80°C, the samples exhibited obvious expansion only caused by ACR, which is beneficial to detect the ACR reactivity of dolomitic rocks exclusively in concrete engineering. Based on the pore structure analysis, there was a slight increase (13%) in porosities of aggregates cured for 140 days at 80°C. Rod-like brucite crystals formed in the process of ACR were also found in TG analysis and SEM images.

# 1. Introduction

Alkali–carbonate reaction (ACR) is a kind of alkali–aggregate reaction, which means the alkali in the concrete reacts with dolomite crystals, leading to map-like cracks in the concrete and deterioration of concrete. Since the detrimental ACR was found by Gillott [1] in the 1960s, extensive research has been carried out regarding the mechanism of the reaction [2–5]. However, the expansion mechanism is still controversial and mainly can be divided into two kinds: one is the growth of brucite crystal causing the expansion [6] and the other is the ACR just like the alkali–silica reaction (ASR) [7–9]. Compared with ASR, the ACR appears to have the following different characteristics: the reaction products are not easy to find due to low amount; the reaction can take place in a low-alkali environment; ASR suppression measures are not suitable for ACR. Currently, although the disagreement still exists, it is widely accepted worldwide that the dolomite in dolomitic limestones interacts with the alkali hydroxides from the concrete pore solution causing a fine intergrowth of calcite and brucite in the alkaline condition. $CO_3^{2-}$ released in the process can migrate to the cement paste dissolving the portlandite phase and releasing $Ca^{2+}$ ions that can react to form secondary calcite and carbonate halo around the aggregate. The solution alkalinity increases due to the regeneration of the alkali hydroxide and the reaction will continue until the dolomite is used up.

Deterioration of concrete in some airports and bridges can be ascribed to the existence of high-alkali cement and aggregates with alkali reactivity in concrete [10]. Although many corresponding methods for detection of potential alkali reactivity of aggregates have been formulated, misjudgement of alkali activity of aggregates also occurs in concrete engineering. Therefore, accurate identification of ACR activity in carbonate aggregates is currently the only effective measure to prevent ACR damage. The current methods for ACR identification mainly include petrographic, rock-prism, concrete microbar and concrete prism methods [11–14]. However, these methods cannot distinguish whether the expansion originates from ACR or ASR or both. Prinčič [15] observed the process of dedolomitization of carbonate aggregate rocks and mortar bars and found the process of ACR occurred not only in the NaOH solution but also in the water on the mortar bar with virgin dolomitic aggregate. Štukovnik [16] indicated that a considerably higher increase in compressive strength was detected over time for the mortar with dolomite aggregate, compared with the one with limestone aggregate, by investigating the process of ACR. But most research is based on NaOH as curing solution, which cannot rule out the expansion that originates from ASR.

Chen *et al.* [17] reported the microcrystalline quartz in dolomitic rocks can hardly react with tetramethyl ammonium hydroxide (TMAH) solution, which can exclude the expansion contribution of ASR. But dolomite in dolomitic rocks can react with TMAH solution and lead to expansion. According to Thong [18], TMAH has been widely used in silicon anisotropic etching due to reliable etching of silicon in the fabrication of microelectromechanical systems. TMAH is used as reactant and $SiO_2$ is the corresponding product in the reaction. So TMAH does not react with $SiO_2$ because reactants and products do not react. Therefore, utilization of TMAH as curing solution can be a specialized and effective way to detect the ACR reactivity of dolomitic rocks. It is necessary to investigate the expansion characteristics and microstructure of dolomitic aggregates before formulating the corresponding standards for judging the ACR reactivity of dolomitic aggregates.

In order to provide a basis for concrete engineering applications in detecting the ACR reactivity of dolomitic aggregates, the modified microbars have been prepared with dolomitic rocks and self-made cement without $K^+$ and $Na^+$. TMAH solution was used as curing solution to investigate the expansion characteristics only caused by ACR under different curing conditions and particle sizes of aggregates. The porosities of original and reacted aggregates were investigated. Additionally, the microstructure of modified microbars cured for 140 days was also investigated according to thermo-gravimetric (TG) analysis and scanning electron microscopy (SEM) images.

# 2. Material and methods

## 2.1. Materials

The cement clinker was prepared by calcining analytic reagent (Xilong Science Co., Ltd), then ground into powders with less than 10% sieve residue. The analytic reagents were directly heated to 1450°C with a heating rate of 10°C min$^{-1}$, with a dwelling time of 1 h at 1450°C. Table 1 shows the raw materials composition of cement clinker without alkali. The reason for the cement without alkali used in the study is ruling out the expansion originates from ASR. Figure 1 shows XRD pattern of cement

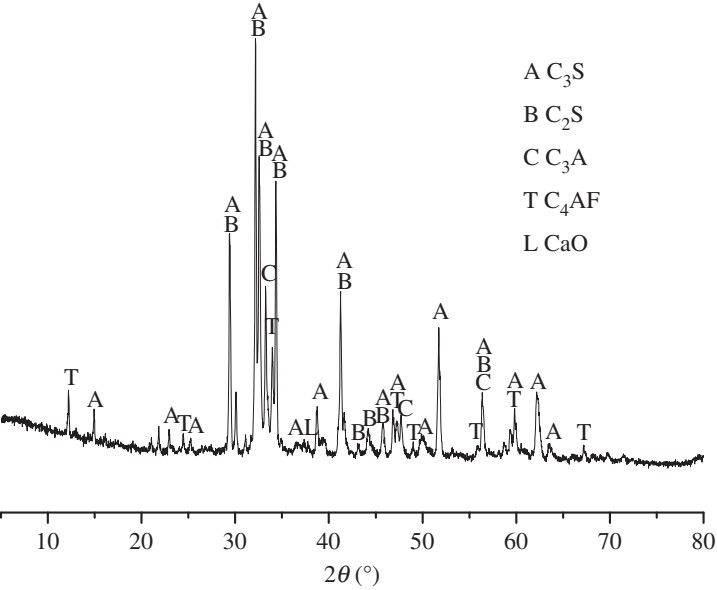

**Figure 1.** XRD pattern of cement clinker without $K^+$ and $Na^+$.

**Table 1.** Raw materials composition of cement clinker without alkali (by weight, %).

| $CaCO_3$ | $SiO_2$ | $Al_2O_3$ | $Fe_3O_4$ | total |
|---|---|---|---|---|
| 78.18 | 14.03 | 4.40 | 3.39 | 100 |

**Table 2.** The mineral contents of cement clinker without $K^+$ and $Na^+$ (%).

| $C_3S$ | $C_2S$ | $C_4AF$ | $C_3A$ | f-CaO | f-MgO |
|---|---|---|---|---|---|
| 64.0 | 11.9 | 10.9 | 13.2 | 0.1 | 0 |

clinker without $K^+$ and $Na^+$. To further characterize the cement clinker, Rietveld analysis was used to investigate the mineral contents of cement clinker without $K^+$ and $Na^+$, as shown in table 2. Gypsum and clinker powders at a weight of 1/19 were blended for 12 h to get homogeneous complete cement without $K^+$ and $Na^+$. The chemical formula of TMAH is $(CH_3)_4NOH$ with 91.15 relative molecular mass. TMAH solution is a colourless and transparent liquid with an ammonia smell. The degree of alkalinity is similar to that of NaOH. Compared with NaOH solution, TMAH solution belongs to organic alkali rather than common inorganic alkali. Dolomitic rocks from Shandong province were used in the study. Figure 2 shows XRD pattern of BFL-1 and BFL-9 rocks. It can be seen that the two rocks were mainly composed of dolomite, calcite and quartz. Table 3 shows chemical composition of the dolomitic rocks used in the study. The content of dolomite in BFL-9 was higher than that in BFL-1. Based on the content of dolomite in the aggregates, they can be divided into dolomitic limestone (BFL-1) and dolostone (BFL-9). According to RILEM AAR-5 [13], samples prepared with BFL-1 and BFL-9 aggregates had an expansion value of 0.136% and 0.237% at 28 days of curing age, respectively, which exhibited the ACR reactivity.

## 2.2. Preparation of modified microbars

BFL-1 and BFL-9 aggregates with different particle size (0–1.25, 1.25–2.5 and 2.5–5 mm) and cement without $K^+$ and $Na^+$ were employed to prepare specimens ($2 \times 2 \times 8$ cm) as the preliminary microbars. The weight ratio of aggregates to cement was fixed to 3/2. For each microbar formulation, aggregates and cement were mixed in a mixer (Type NJ-160A, Shanghai Luheng Co. Ltd, China). Samples completely made of cement were used as the reference specimens. All specimens with the

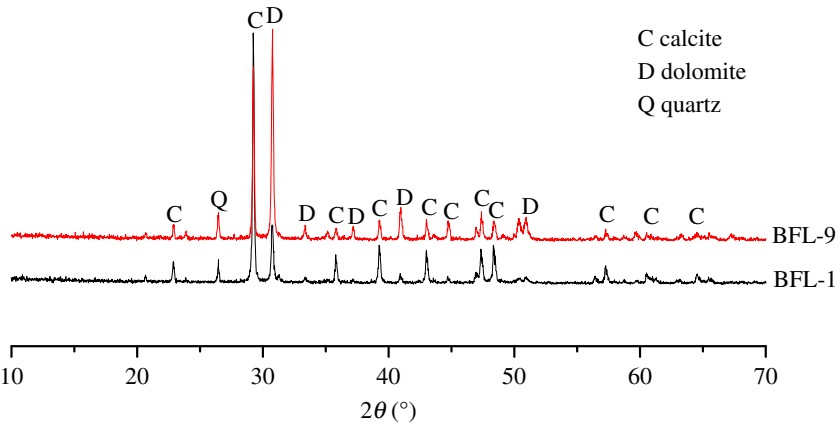

**Figure 2.** XRD pattern of BFL-1 and BFL-9 rocks.

**Table 3.** Chemical composition of the aggregates used in the study.

| samples | chemical compositions (wt%) | | | | | | | | | |
|---|---|---|---|---|---|---|---|---|---|---|
| | LOI[a] | $SiO_2$ | CaO | MgO | $Al_2O_3$ | $Fe_2O_3$ | $SO_3$ | $K_2O$ | $Na_2O$ | total |
| BFL-1 | 42.05 | 4.70 | 43.61 | 6.35 | 1.23 | 0.50 | 0.06 | 0.61 | 0.09 | 99.20 |
| BFL-9 | 41.26 | 5.97 | 37.14 | 12.09 | 1.13 | 0.63 | 0.07 | 0.48 | 0.10 | 98.87 |

[a]Loss on ignition.

mould were placed in a moist environment (RH = 98%) at 20°C. After 24 h, the specimens were taken from the mould and modified by cutting machine to obtain the modified microbars with 8 cm length, 2 cm width and 5 mm thickness. Then the initial length measurement was taken and the bars were transferred to containers filled with TMAH solutions (Zhenjiang Runjing Co. Ltd, China). There are two reasons for the modification. One is to reduce the self-shrinkage of the cement, and the other is to accelerate the ACR by directly contacting the aggregate with the alkali solution and then contribute to more obvious expansion. Curing conditions for the specimens made with the aggregates with ACR activity were: $t = 20$, 60 and 80°C, and $c = 1$, 2 N (concentration of TMAH).

## 2.3. Testing and characterization

X-ray diffraction (Smart Lab, Rigaku, Tokyo, Japan) analysis was used for the composition of self-made cement without $K^+$ and $Na^+$ and the mineralogical detection of dolomitic rocks. The length changes of all the specimens prepared by BFL-1 and BFL-9 aggregates were measured at different intervals and the expansions of modified microbars were calculated by JC/T 313–2009 (Chinese Standard). Each length change value used was the mean value of five replicate specimens. The porosities and pore size distribution of original and reacted aggregates were tested by mercury intrusion porosimetry (MIP). Corresponding thermo-gravimetric analysis was carried out by thermal gravimetric analyser (SDT Q600) at temperatures ranging from 20 to 950°C with $N_2$ ambience. The morphologies of BFL-9 grains enriched dolomite selected from the modified microbars cured for 140 days were also observed by scanning electron microscopy (SEM, JEOL, JEM-6510) coupled with energy dispersive spectrometer (EDS) analysis.

# 3. Results and discussion

## 3.1. Effect of aggregate grain size on the expansion of modified microbars cured in TMAH solution

Changes in particle size of aggregate not only influence the microbar structure, but also the ion migration during the process of ACR [19]. To investigate effect of particle size of aggregate on the expansion of modified microbars, aggregates with 0–1.25, 1.25–2.5 and 2.5–5 mm particle size were used to prepare

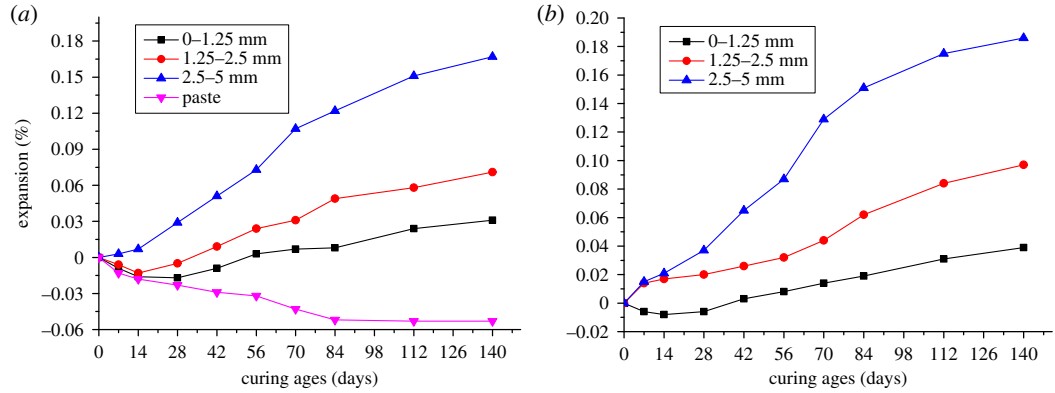

**Figure 3.** The expansion of modified microbars prepared by BFL-1 (*a*) and BFL-9 (*b*) rocks with different particle size.

microbars cured at 80°C in 1 N TMAH solution. Figure 3 shows the expansion of microbars prepared by BFL-1 and BFL-9 rocks. From figure 3*a*, the expansion process of microbars can be divided into three stages. For the first stage, the microbars prepared by aggregates with different particle size appear slow expansion. The expansion of bars at 14 days was just 0.007%. Compared with microbars made by 2.5–5 mm aggregates, bars made by finer aggregates and paste had shrinkage at early curing ages. The shrinkage of paste was 0.013%. For the second stage, bars had obvious expansion as curing ages increase and the differences in expansion began to expand. But the paste still had obvious shrinkage. For the last stage, in addition to the paste, the expansion of bars appears slow again and tends to be stable. At 140 days, the shrinkage of paste was 0.053%, but the expansion of bars was up to 0.167%. From figure 3*b*, the expansion process of bars was similar to that in figure 3*a*. Compared with BFL-1, the samples made with BFL-9 had bigger expansion. The maximum expansion of BFL-9 bars was 0.186%.

At early curing ages, samples with slow expansion can be ascribed to the obvious shrinkage of cement without alkali, and the shrinkage offset some expansion originated from ACR. Cement shrinkage can be ascribed to the formation of hydration products with lower volume during the early ages. From table 3, it can be seen that the content of dolomite in BFL-9 was higher than that in BFL-1, which may account for bigger expansion. As shown in figure 3, the differences in the expansion of bars made by different particle size were obvious. The differences can be explained by that, when the aggregate size and cement particle size are similar, the expansion of the bars is more susceptible to the shrinkage of the cement. From the results in figure 3, it was indicated that particle size of the aggregates can influence the expansion of microbars significantly. Compared with 0–1.25 and 1.25–2.5 mm, microbars prepared by 2.5–5 mm aggregates had bigger expansion and 2.5–5 mm was more suitable as the particle size. In addition, the expansions of samples cured in TMAH solution were less than the expansions of the same samples cured in the NaOH solution due to excluding the expansion contribution of ASR. The expansions of samples cured in TMAH solution developed more slowly than that cured in NaOH at early age.

## 3.2. Effect of curing temperature on the expansion of modified microbars cured in TMAH solution

In addition to the particle size of aggregates, curing temperature also affects the expansion of modified microbars. Specimens prepared by aggregates with 2.5–5 mm particle size were cured in 1 mol l⁻¹ TMAH solution at 20, 60 and 80°C, respectively. Figure 4*a* shows the expansion of microbars made by BFL-1 rocks cured at 20, 60 and 80°C. For BFL-1 bars, the differences in expansion between bars cured at different temperature were small. But the differences became obvious after 14 days. When the curing age was 140 days, the expansions of bars cured at 20, 60 and 80°C were 0.025%, 0.115% and 0.167%, respectively, and tended to be stable. When the curing temperature increased from 60°C to 80°C, the expansion of bars cured at 80°C was 45% larger than that cured at 60°C. From figure 4*b*, compared with BFL-1 rocks, the differences in the expansion of BFL-9 were more obvious at early curing age. The whole expansion process was similar to that of BFL-1 and the bars had slow expansion after 112 days. The expansions of bars cured at 20, 60 and 80°C were 0.043%, 0.125% and 0.186% at 140 days curing age. For BFL-9 aggregates, the expansion of microbar cured at 80°C was 49% larger than that of microbar cured at 60°C. When the curing temperature increased, the expansion of microbars became larger. Differences in the expansion of microbars cured at different

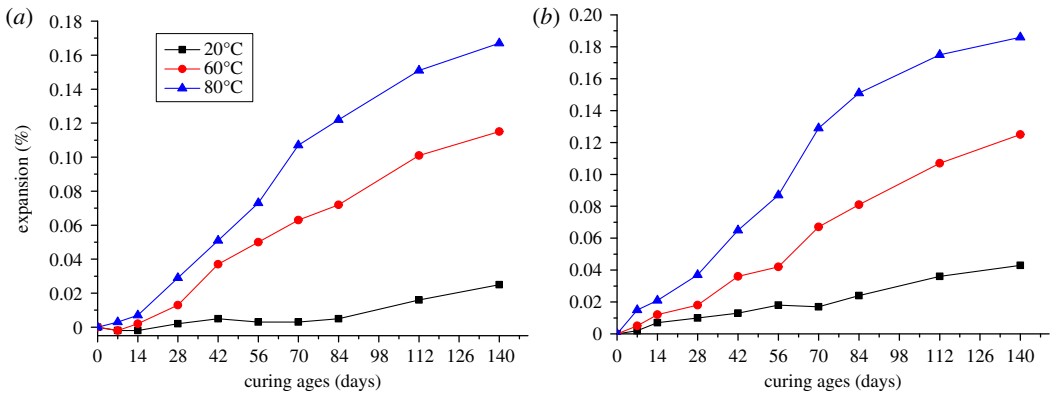

**Figure 4.** The expansion of modified microbars prepared by BFL-1 (*a*) and BFL-9 (*b*) rocks cured at 20, 60 and 80°C.

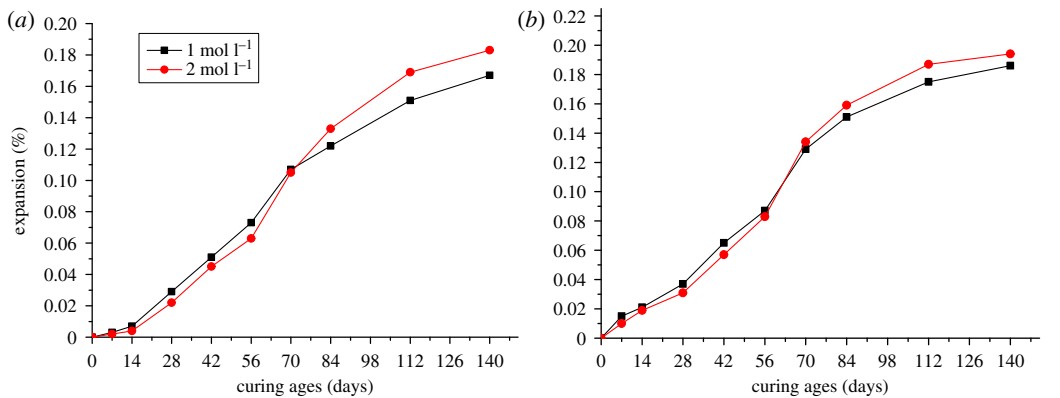

**Figure 5.** The expansion of modified microbars prepared by BFL-1 (*a*) and BFL-9 (*b*) rocks cured in 1 and 2 mol l$^{-1}$ TMAH solutions.

temperatures can be ascribed to the reaction rate of ACR. Increasing the curing temperature improved the rate of alkaline ion migration into the interior of dolomite and accelerated the ACR, thus promoting the expansion of modified microbars. Therefore, selecting 80°C as the curing temperature can shorten the time for experiment.

## 3.3. Effect of concentration of TMAH solution on the expansion of modified microbars

To further investigate the influence of the concentration of curing solution on the expansion of modified microbars, the samples prepared by BFL-1 and BFL-9 aggregates with 2.5–5 mm particle size at 80°C were cured in 1 and 2 mol l$^{-1}$ TMAH solutions, respectively. The results are given in figure 5. From figure 5*a*, samples made by BFL-1 aggregates had small expansion at early curing age. After 14 days, the corresponding expansion became obvious. There was a shape increase in 56 days curing age. But the expansion of samples became stable after 112 days. It can be seen that the differences in expansion between samples cured in 1 and 2 mol l$^{-1}$ were slight. Compared with cured in 1 mol l$^{-1}$ TMAH solution, samples cured in 2 mol l$^{-1}$ had slight lower expansion before 70 days, but higher expansion appeared after 70 days. The expansion of BFL-1 samples cured in 1 and 2 mol l$^{-1}$ TMAH solutions were 0.167% and 0.183% at 140 days. As shown in figure 5*b*, the expansion process of samples made by BFL-9 aggregates was just like BFL-1 aggregates, except bigger expansion. The expansions of BFL-9 samples cured in 1 and 2 mol l$^{-1}$ TMAH solutions were 0.186% and 0.194% at 140 days. No obvious differences between samples cured in 1 and 2 mol l$^{-1}$ can be ascribed to the sufficient alkaline in these two concentrations of alkali solutions. The 1 mol l$^{-1}$ TMAH solution may provide enough alkaline in the process of ACR, but at later curing age, the alkaline may not be enough for the alkali-consuming ACR, which may account for samples cured in 2 mol l$^{-1}$ TMAH solution with higher expansion than that cured in 1 mol l$^{-1}$ TMAH solution. By taking expansion characteristics as well as economy into consideration, the particle size, curing temperature and concentration of TMAH solution were fixed at 2.5–5 mm, 80°C and 1 mol l$^{-1}$, respectively, to investigate the microstructure of dolomitic aggregates cured in TMAH solution.

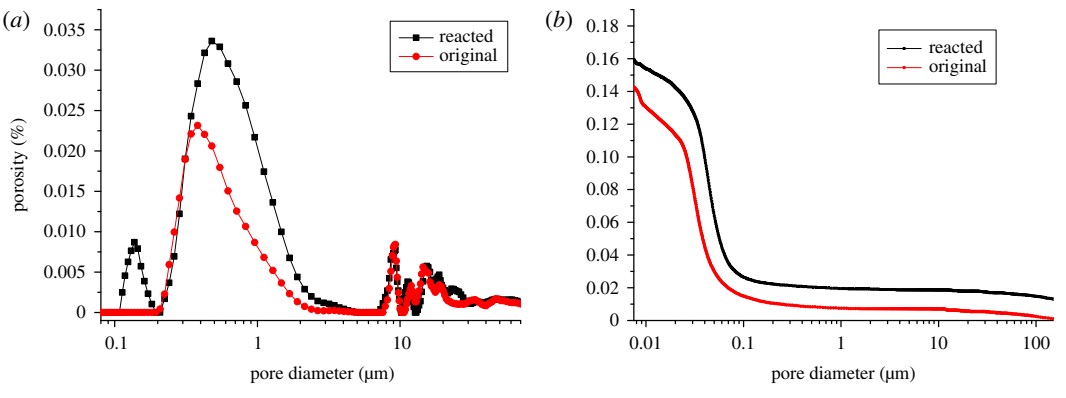

**Figure 6.** Typical pore structure of BFL-9 aggregates: (*a*) derivative porosity curve (pore size distribution) and (*b*) cumulative porosity curve.

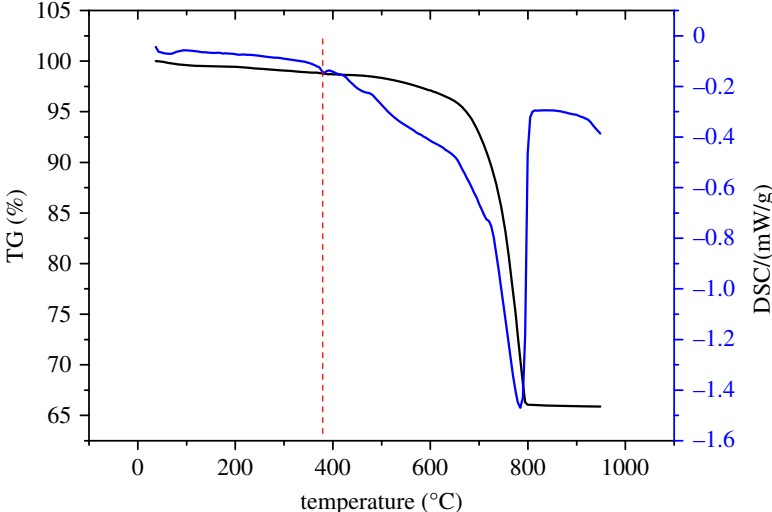

**Figure 7.** DTA-TG curves of BFL-9 dolomitic aggregate grain cured in TMAH solution for 140 days.

## 3.4. Microstructure

In order to investigate the effect of ACR on the pore structures of dolomitic rocks, figure 6 shows the pore size distribution of BFL-9 original aggregate grains and reacted aggregate grains cured in TMAH solution for 140 days. It can be seen that the pores in original aggregates have a pore diameter range of 0.2–1.1 µm, but the pores in reacted aggregates have a pore diameter range of 0.1–1.2 µm. Compared with the original aggregate grain in pore size distribution, the reacted aggregate grain had new-formed micro-pores with pore diameter ranging from 0.1 to 0.2 µm. As shown in figure 6*b*, the total porosities of original and reacted aggregates were 0.143% and 0.162%, respectively. Compared with the original aggregates in porosity, the reacted aggregates exhibited a slight increase (13%) in total porosities at 140 days, which was similar to the results from Štukovnik [16]. This may be due to that the growth of brucite crystal as a result of ACR in the restricted space dilated the pores.

To characterize the reaction products of ACR, figure 7 shows TG-DSC analysis on BFL-9 dolomitic grain cured in TMAH solution for 140 days. It can be seen that there was a slight peak at 380°C, which represented the existence of brucite in reacted dolomitic aggregates. The endothermic peak appearing at the DSC curves, namely from 350°C to 400°C, corresponds to the dehydration of $Mg(OH)_2$. To quantify the amounts of the brucite, the contents of $Mg(OH)_2$ contained in the aggregate grain could be estimated according to the following equation:

$$mass_{Mg(OH)_2} = \frac{58 \times mass\,loss(350°C - 400°C)}{18},$$

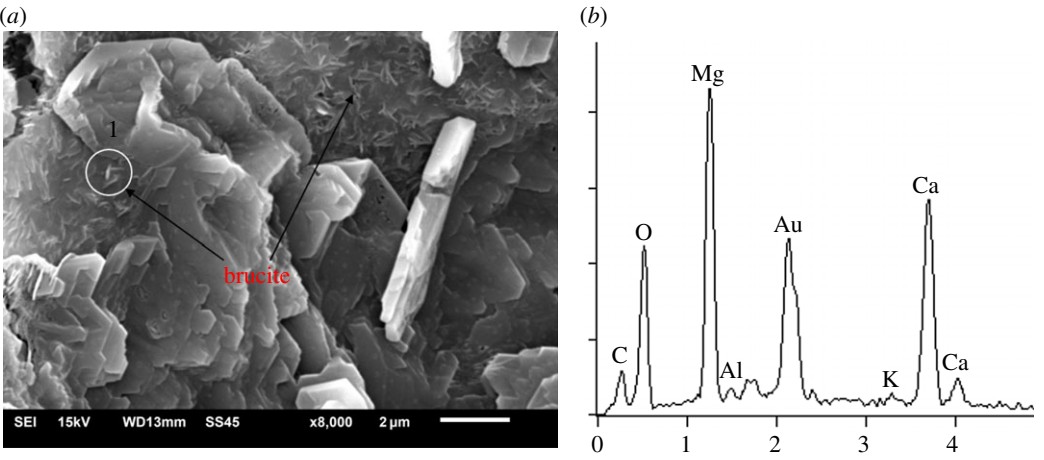

**Figure 8.** SEM-EDS analysis. (*a*) SEM image of BFL-9 aggregate with 2.5–5 mm grain cured in TMAH at 80°C for 140 days. (*b*) EDS spectra determined in zone 1.

where the $mass_{Mg(OH)_2}$ refers to the contents of $Mg(OH)_2$ formed in the process of ACR, and 58 and 18 represent the relative molecular mass of $Mg(OH)_2$ and $H_2O$, respectively. On the basis of the above-mentioned equation, the amounts of the brucite contained in BFL-9 dolomitic grain were 0.61%. The slight peak can be ascribed to the small amount of brucite. According to Tong [2], brucite crystals grow up in confined spaces, bringing about the expansion of dolomitic aggregates, which may account for the changes in total porosities. To further characterize brucite formed in the process of ACR, SEM-EDS analysis of dolomitic aggregate grains was carried out.

From Katayama [8,9], according to polarizing microscopy, it was observed that ACR produced a myrmekitic texture, which was composed of spotted brucite and calcite within the reaction rim. But to further investigate the reaction products of ACR, as shown in figure 8, the reacted BFL-9 aggregate with dolomite enrichment area and cracks was selected to be observed by SEM-EDS analysis. From figure 8a, it can be seen that there are many rod-like crystals beside the micro-crack. Figure 8b shows EDS spectra determined in zone 1. Combining the two figures, it is indicated that rod-like brucite crystals formed in the process of ACR indeed exist. Brucite and calcite were the reaction products result from dolomite reacting with alkali in the aggregate grain. It should be noted that unreacted dolomite still exists in the aggregate grain because some dolomite has not yet fully reacted in the process of dedolomitization. When the pH and temperature of the curing solution is high enough, the ACR can continue to occur. Owing to TMAH solution used in this work to rule out the influence from microcrystalline quartz and exclude the expansion contribution of ASR, the expansion only originates from the ACR. Therefore, the expansion of the modified microbars cured in TMAH solutions can be ascribed to the existence and growth of brucite crystals in confined space. Based on the aforementioned discussion, it can be indicated that the ACR can occur in some special dolomitic aggregates cured in 1 mol l$^{-1}$ TMAH solution at 80°C and brings about expansion.

## 4. Conclusion

In this work, the modified microbars prepared by dolomitic aggregates and self-made cement were used to systematically investigate effects of the concentration of TMAH solution, curing temperature and aggregate grain size on the expansion of dolomitic aggregates. From the physical measurement and microstructural analysis, the following major conclusions can be drawn.

Changing the aggregate grain size for modified microbars and increasing the curing temperature can considerably increase the rate of ACR and then accelerate the samples with bigger expansion. Compared with 0–1.25 and 1.25–2.5 mm particle size, modified microbars prepared with 2.5–5 mm grains have bigger expansion. The expansion of microbars cured at 80°C is 49% larger than that cured at 60°C for 140 days, much larger than that cured at 20°C. The expansion of microbars cured in 1 and 2 mol l$^{-1}$ TMAH solution have no obvious differences, which is probably due to adequate alkaline for ACR.

Based on the analysis of pore size distribution and total porosities of aggregate grains, the porosity increased 13% as result of ACR. According to the TG analysis assisted SEM-EDS analysis, it was revealed that rod-like brucite crystals formed in the process of ACR were found.

Due to TMAH solution used in this work to rule out the influence from microcrystalline quartz and exclude the expansion contribution of ASR, the expansion only origins from the ACR. The ACR can occur in some special dolomitic aggregates cured in $1 \, \text{mol} \, l^{-1}$ TMAH solution at $80°C$ and brings about expansion.

Data accessibility. All data are included in the article.

Authors' contributions. B.Y. designed and conducted the experimental program, and drafted the manuscript. X.C. carried out the statistical analyses and collected field data. M.D. provided and designed the project. All authors contributed to the analysis and conclusion. And all authors gave final approval for publication.

Competing interests. We declare we have no competing interests.

Funding. This work was supported by the National Key Research and Development Plan of China (2017YFB0309903-01) and the Priority Academic Program Development of Jiangsu Higher Education Institutions (PAPD).

Acknowledgements. The authors gratefully acknowledge the assistance from Mr Liu Peng and Mr Yu Qing from NJTECH, and the staff from State Key Laboratory of Materials-Oriented Chemical Engineering.

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
