## [Reviewer comments · Royal Society Open Science]

Review History

RSOS-190044.R0 (Original submission)

Review form: Reviewer 1

Is the manuscript scientifically sound in its present form?

Yes

Are the interpretations and conclusions justified by the results?

Yes

Is the language acceptable?

Yes

Is it clear how to access all supporting data?

Yes

Do you have any ethical concerns with this paper?

No

Have you any concerns about statistical analyses in this paper?

No

Recommendation?

Major revision is needed (please make suggestions in comments)

Comments to the Author(s)

This is an interesting work that discusses TMAH solution was utilized as curing solution to investigate the expansion characteristics only caused by ADR. However, several aspects of the work must be improved before it can be accepted for publication. I have listed them in detail below:

1. Details of TMAH solution are not presented.

What are the similarities and differences between TMAH solution and NaOH solutions? such as PH? chemical formula?

2. Most research about AAR(alkali-aggregate reaction) is based on NaOH solution, which cannot rule out the expansion originates from ASR or ACR. But TMAH can exclude the expansion contribution of ASR. What is the effect mechanism? What are the differences between TMAH and NaOH resulted in the different expansion?

3. Table 1 the composition of cement clinker, CaCO_3 should be changed to CaO .

4. Page10Line31, which represented the existence of brucite in reacted dolomitic aggregates. Can you quantify the amounts of the brucite?

Review form: Reviewer 2**Is the manuscript scientifically sound in its present form?**

Yes

Are the interpretations and conclusions justified by the results?

Yes

Is the language acceptable?

Yes

Is it clear how to access all supporting data?

Not Applicable

Do you have any ethical concerns with this paper?

No

Have you any concerns about statistical analyses in this paper?

I do not feel qualified to assess the statistics

Recommendation?

Accept with minor revision (please list in comments)

Comments to the Author(s)

The comments are detailed in the pdf file (Appendix A).

Decision letter (RSOS-190044.R0)

25-Feb-2019

Dear Dr Yang:

Title: Effects of curing condition and particle size of aggregate on the expansion and microstructure of dolomitic aggregates cured in TMAH solution
Manuscript ID: RSOS-190044

The editor assigned to your manuscript has now received comments from reviewers. We would like you to revise your paper in accordance with the referee and Subject Editor suggestions which can be found below (not including confidential reports to the Editor). Please note this decision does not guarantee eventual acceptance.

Please submit your revised paper before 20-Mar-2019. Please note that the revision deadline will expire at 00.00am on this date. If we do not hear from you within this time then it will be assumed that the paper has been withdrawn. In exceptional circumstances, extensions may be possible if agreed with the Editorial Office in advance. We do not allow multiple rounds of revision so we urge you to make every effort to fully address all of the comments at this stage. If deemed necessary by the Editors, your manuscript will be sent back to one or more of the original reviewers for assessment. If the original reviewers are not available we may invite new reviewers.

RSC Associate Editor:
Comments to the Author:
(There are no comments.)

RSC Subject Editor:
Comments to the Author:
(There are no comments.)

Reviewers' Comments to Author:
Reviewer: 1

Comments to the Author(s)

This is an interesting work that discusses TMAH solution was utilized as curing solution to investigate the expansion characteristics only caused by ADR. However, several aspects of the work must be improved before it can be accepted for publication. I have listed them in detail below:

1. Details of TMAH solution are not presented.

What are the similarities and differences between TMAH solution and NaOH solutions? such as PH? chemical formula?

2. Most research about AAR(alkali-aggregate reaction) is based on NaOH solution, which cannot rule out the expansion originates from ASR or ACR. But TMAH can exclude the expansion contribution of ASR. What is the effect mechanism? What are the differences between TMAH and NaOH resulted in the different expansion?

3. Table 1 the composition of cement clinker, CaCO_3 should be changed to CaO .

4. Page10Line31, which represented the existence of brucite in reacted dolomitic aggregates. Can you quantify the amounts of the brucite?

Reviewer: 2

Comments to the Author(s)
The comments are detailed in the pdf file

Author's Response to Decision Letter for (RSOS-190044.R0)

See Appendix B.

Decision letter (RSOS-190044.R1)

19-Mar-2019

Dear Dr Yang:

Title: Effects of curing condition and particle size of aggregate on the expansion and microstructure of dolomitic aggregates cured in TMAH solution

Manuscript ID: RSOS-190044.R1

It is a pleasure to accept your manuscript in its current form for publication in Royal Society Open Science. The chemistry content of Royal Society Open Science is published in collaboration with the Royal Society of Chemistry.

RSC Associate Editor
Comments to the Author:
(There are no comments.)

Reviewer(s)' Comments to Author:

Appendix A**ROYAL SOCIETY
OPEN SCIENCE****Effects of curing condition and particle size of aggregate on
the expansion and microstructure of dolomitic aggregates
cured in TMAH solution**

Journal:	Royal Society Open Science
Manuscript ID	RSOS-190044
Article Type:	Research
Date Submitted by the Author:	17-Jan-2019
Complete List of Authors:	Yang, Bin; Nanjing Tech University Chen, Xiaoxiao; Nanjing Tech University Deng, Min; Nanjing Tech University
Subject:	Materials science < CHEMISTRY
Keywords:	Alkali-dolomite reaction, Dolomitic aggregates, Expansion, Microstructure
Subject Category:	Chemistry

Effects of curing condition and particle size of aggregate on the expansion and microstructure of dolomitic aggregates cured in TMAH solution

Bin Yang, Xiaoxiao Chen, Min Deng*

College of Materials science and Engineering, Nanjing Tech University, Nanjing, China

Keywords: Alkali-dolomite reaction; Dolomitic aggregates; Expansion; Microstructure

Abstract

In this paper, the modified microbars prepared by dolomitic aggregates with three kinds of particle size and self-made cement without K^+ , Na^+ were cured in 1 and 2 N tetramethyl ammonium hydroxide (TMAH) solution at 20, 60 and 80 °C, respectively. TMAH was utilized as curing solution to exclude the expansion contribution of alkali-silica reaction (ASR). Effects of the concentration of TMAH solution, curing temperature and aggregate grain size on the expansion of dolomitic aggregates were systematically investigated to determine the expansion characteristics only caused by alkali-dolomite reaction (ADR). Expansion of modified microbars cured in TMAH solution were measured. The porosities of original and reacted aggregates were also measured. Microstructural studies were carried out by scanning electron microscopy (SEM) and thermo-gravimetric (TG) analysis. The results showed that the aggregate grain size and curing temperature can influence the expansion of modified microbars significantly. When the modified microbars prepared by aggregates with 2.5-5 mm grain size and cured in 1 N TMAH solution at 80 °C, the samples exhibited obvious expansion only caused by ADR, which is beneficial to detect the ADR reactivity of dolomitic rocks exclusively in concrete engineering. Based on the pore structure analysis, there was a slight increase (13%) in porosities of aggregates cured for 140 d at 80 °C. Rod-like brucite crystals formed in the process of ADR were also found in TG analysis and SEM images.

Introduction

Alkali-carbonate reaction (ACR) also called alkali-dolomite reaction (ADR) is one of important kinds of alkali-aggregate reaction, which means the alkali in the

1
2
3
4
5
6
7
8
9
10
11
12
13
14
15
16
17
18
19
20
21
22
23
24
25
26
27
28
29
30
31
32
33
34
35
36
37
38
39
40
41
42
43
44
45
46
47
48
49
50
51
52
53
54
55
56
57
58
59
60

concrete reacts with dolomite crystals, leading to map-like cracks in the concrete and deterioration of concrete. Since the detrimental ACR was found by Gillott [1] in the 1963s, extensive research has been carried out regarding the mechanism of the reaction [2-5]. However, the concrete mechanism is still controversial and mainly can be divided into two kinds: one is the growth of brucite crystal causing the expansion [6], the other is the ACR just like to alkali-silica reaction (ASR) [7-9]. Compared to ASR, the ADR appears the following different characteristics: the reaction products are not easy to find due to low amount; the reaction can take place in a low alkali environment; ASR suppression measures are not suitable for ADR. Currently, although the disagreement still exists, it is widely accepted worldwide that the dolomite in dolomitic limestones interacts with the alkali hydroxides from the concrete pore solution causing a fine intergrowth of calcite and brucite in the alkaline condition. CO_3^{2-} released in the process can migrate to the cement paste dissolving the portlandite phase and releasing Ca^{2+} ions that can react to form secondary calcite and carbonate halo around the aggregate. The solution alkalinity increases due to the regeneration of the alkali hydroxide and the reaction will continue until the dolomite is used up.

Deterioration of concrete in some airports and bridges can be ascribed to the existence of high-alkali cement and aggregates with alkali-reactivity in concrete [10]. Although many corresponding methods for detection of potential alkali-reactivity of aggregates have been formulated, misjudgment of alkali-activity of aggregates also occurs in concrete engineering. Therefore, accurate identification of ADR activity in carbonate aggregates is currently the only effective measure to prevent ADR damage. The current methods for ADR identification mainly include petrographic method, rock-prism method, concrete microbar and concrete prism method [11-14]. However, these methods cannot distinguish whether the expansion originates from ADR or ASR or both. Tina Prinčič [15] observed the process of dedolomitization of carbonate aggregate rocks and mortar bars and found the process of ADR occurred not only in the NaOH solution but also in the water on the mortar bar with virgin dolomitic aggregate. P. Stukovnik [16] indicated that a considerably higher increase in compressive strength was detected over time for the mortar with dolomite aggregate, compared to the one with limestone aggregate by investigating the process of ACR. But most research is based on NaOH as curing solution, which cannot rule out the expansion originates from ASR.

Chen [17] reported the microcrystalline quartz in dolomitic rocks can hardly react with tetramethyl ammonium hydroxide (TMAH) solution, which can exclude the expansion contribution of ASR. But dolomite in dolomitic rocks can react with TMAH solution and lead to expansion. Therefore, utilization of TMAH as curing solution can be a specialized and effective way to detect the ADR reactivity of dolomitic rocks. It is necessary to investigate the expansion characteristics and microstructure of dolomitic aggregates before formulating the corresponding standards for judging the ADR reactivity of dolomitic aggregates.

In order to provide a basis for concrete engineering applications in detecting the ADR reactivity of dolomitic aggregates, the modified microbars has been prepared by

dolomitic rocks and self-made cement without K^+ , Na^+ . TMAH solution was utilized as curing solution to investigate the expansion characteristics only caused by ASR under different curing conditions and particle sizes of aggregates. The porosities of original and reacted aggregates were investigated. Additionally, the microstructure of modified microbars cured for 140 d was also investigated according to TG analysis and SEM images.

Materials and Methods

Materials

The cement clinker was prepared by calcining analytic reagent (Xilong Science Co., Ltd.), followed by ground into powders with less than 10% sieve residue. The analytic reagents were directly heated to 1450 °C with a heating rate of 10 °C /min, with a dwelling time of 1 hour at 1450 °C. Table 1 shows the raw materials composition of cement clinker without alkali. The reason for the cement without alkali used in the study is ruling out the expansion originates from ASR. Figure 1 shows XRD pattern of cement clinker without K^+ and Na^+ . To further characterize the cement clinker, rietveld analysis was utilized to investigate the mineral contents of cement clinker without K^+ and Na^+ , as shown in Table 2. Gypsum and clinker powders at a weight of 1/19 were blended for 12 h to get homogenous complete cement without K^+ , Na^+ . Dolomitic rocks from Shandong province were used in the study. Figure 2 shows XRD pattern of BFL-1 and BFL-9 rocks. It can be seen that the two rocks were mainly composed of dolomite, calcite and a little amount of quartz. Table 3 shows chemical composition of the dolomitic rocks used in the study. The content of dolomite in BFL-9 was higher than that in BFL-1. Based on the content of dolomite in the aggregates, they can be divided into dolomitic limestone (BFL-1) and dolostone (BFL-9). According to RILEM AAR-5 [13], samples prepared by BFL-1, BFL-9 aggregates had expansion value of 0.136% and 0.237% at 28 days of curing age, respectively, which exhibited the ACR reactivity.

Table 1: Raw materials composition of cement clinker without alkali (by weight, %).

$CaCO_3$	SiO_2	Al_2O_3	Fe_3O_4	Total
78.18	14.03	4.40	3.39	100

Table 2: The mineral contents of cement clinker without K^+ , Na^+ (%).

C_3S	C_2S	C_4AF	C_3A	f-CaO	f-MgO
64.0	11.9	10.9	13.2	0.1	0

Figure 1: XRD pattern of cement clinker without K⁺, Na⁺.

Figure 2: XRD pattern of BFL-1 and BFL-9 rocks.

Table 3: Chemical composition of the aggregates used in the study.

Samples	Chemical compositions /wt. %									
	LOI*	SiO ₂	CaO	MgO	Al ₂ O ₃	Fe ₂ O ₃	SO ₃	K ₂ O	Na ₂ O	Total
BFL-1	42.05	4.70	43.61	6.35	1.23	0.50	0.06	0.61	0.09	99.20
BFL-9	41.26	5.97	37.14	12.09	1.13	0.63	0.07	0.48	0.10	98.87

*Loss on ignition

Preparation of modified microbars

BFL-1 and BFL-9 aggregates with different particle size (0-1.25, 1.25-2.5, 2.5-5 mm) and cement without K⁺, Na⁺ were employed to prepare specimens (2×2×8 cm) as the preliminary microbars. The weight ratio of aggregates to cement was fixed to 3/2. For each microbar formulation, aggregates and cement were mixed in a mixer (Type NJ-160A, Shanghai Luheng Co. Ltd, China.). Samples completely made by cement were used as the reference specimens. All specimens with the mould were placed in a moist environment (RH=98%) at 20 °C. After 24 h the specimens were taken from the

1
2
3 mould and modified by cutting machine to obtain the modified microbars with 8 cm
4 length, 2 cm width and 5 mm thickness. Then the initial length measurement was
5 taken and the bars were transferred to containers filled with tetramethyl ammonium
6 hydroxide solutions (Zhenjiang Runjing Co. Ltd, China.). There are two reasons for
7 the modification. One is to reduce the self-shrinkage of the cement, and the other is to
8 accelerate the alkali-dolomite reaction by directly contacting the aggregate with the
9 alkali solution and then contribute to more obvious expansion. Curing condition for
10 the specimens made by the aggregates with ACR activity were: $t=20, 60$ and $80\text{ }^{\circ}\text{C}$,
11 and $c=1, 2\text{ N}$ (concentration of TMAH).
12
13
14
15

16 **Testing and characterization**

17
18
19 X-ray diffraction (Smart Lab, Rigaku, Tokyo, Japan) analysis was used for the
20 composition of self-made cement without K^+ , Na^+ and the mineralogical detection of
21 dolomitic rocks. The length changes of all the specimens prepared by BFL-1 and
22 BFL-9 aggregates were measured at different intervals and the expansions of
23 modified microbars were calculated by JC/T 313-2009 (Chinese Standard). Each
24 length change value used was the mean value of five replicate specimens. The
25 porosities and pore size distribution of original and reacted aggregates were tested by
26 mercury intrusion porosimetry (MIP). Corresponding thermo-gravimetric analysis
27 was carried out by thermal gravimetric analyzer (SDT Q600) at temperature ranging
28 from 20 to $950\text{ }^{\circ}\text{C}$ with N_2 ambience. The morphologies of BFL-9 grains enriched
29 dolomite selected from the modified microbars cured for 140 days were also observed
30 by scanning electron microscopy (SEM, JEOL, JEM-6510) coupled with energy
31 dispersive spectrometer (EDS) analysis.
32
33
34
35
36
37

38 **Results and Discussion**

39 **Effect of aggregate grain size on the expansion of modified microbars cured in**

40 **TMAH solution**

41
42
43
44
45
46
47 Changes in particle size of aggregate not only influence the microbar structure
48 but also the ion migration during the process of alkali-dolomite reaction [18]. To
49 investigate effect of particle size of aggregate on the expansion of modified microbars,
50 aggregates with 0-1.25, 1.25-2.5, 2.5-5 mm particle size were used to prepare
51 microbars cured at $80\text{ }^{\circ}\text{C}$ in 1 N TMAH solution. Figure 3 shows the expansion of
52 microbars prepared by BFL-1 and BFL-9 rocks. From Figure 3(a), the expansion
53 process of microbars can be divided into three stages. For the first stage, the
54 microbars prepared by aggregates with different particle size appear slow expansion.
55 The expansion of bars at 14 d was just 0.007%. Compared to microbars made by 2.5-5
56 mm aggregates, bars made by finer aggregates and paste had shrinkage at early curing
57 ages. The shrinkage of paste was 0.013%. For the second stage, bars had obvious
58
59
60

expansion as curing ages increase and the differences in expansion began to expand. But the paste still had obvious shrinkage. For the last stage, in addition to the paste, the expansion of bars appear slow again and tend to stable. At 140 d, the shrinkage of paste was 0.053%, but the expansion of bars was up to 0.167%. From Figure 3(b), the expansion process of bars was similar to that in Figure 3(a). Compare to BFL-1, the samples made by BFL-9 had bigger expansion. The maximum expansion of BFL-9 bars was 0.186%.

At early curing ages, samples with slow expansion can be ascribed to the obvious shrinkage of cement without alkali and the shrinkage offset some expansion originated from AD₃₃ Cement shrinkage can be ascribed to the formation of hydration products with lower volume during the early ages. The whole expansion of microbars was not quite big due to TMAH solution ruling out the expansion came from ASR. From Table 3, it can be seen that the content of dolomite in BFL-9 was higher than that in BFL-1, which may account for bigger expansion. As shown in Figure 3, the differences in the expansion of bars made by different particle size were obvious. The differences can be explained by that when the aggregate size and cement particle size are similar, the expansion of the bars is more susceptible to the shrinkage of the cement. From the results in Figure 3, it was indicated that particle size of the aggregates can influence the expansion of microbars significantly. Compared with 0-1.25, 1.25-2.5 mm, microbars prepared by 2.5-5 mm aggregates had bigger expansion and 2.5-5 mm was more suitable as the particle size.

Figure 3: The expansion of modified microbars prepared by BFL-1 (a) and BFL-9 (b) rocks with different particle size.

Effect of curing temperature on the expansion of modified microbars cured in

TMAH solution

In addition to the particle size of aggregates, curing temperature also affects the expansion of modified microbars. Specimens prepared by aggregates with 2.5-5 mm particle size were cured in 1 mol/L TMAH solution at 20, 60 and 80 °C, respectively. Figure 4 (a) shows the expansion of microbars made by BFL-1 rocks cured at 20, 60

and 80 °C. For BFL-1 bars, the differences in expansion between bars cured at different temperature were small. But the differences became obvious after 14 days. When the curing age was 140 d, the expansions of bars cured at 20, 60 and 80 °C were 0.025%, 0.115% and 0.167%, respectively and tend to stable. When the curing temperature increased from 60°C to 80°C, the expansion of bars cured at 80 °C was 45% larger than that cured at 60 °C. From Figure 4 (b), compared to BFL-1 rocks, the differences in the expansion of BFL-9 were more obvious at early curing age. The whole expansion process was similar to that of BFL-1 and the bars had slow expansion after 112 d. The expansions of bars cured at 20, 60 and 80 °C were 0.043%, 0.125% and 0.186% at 140 d curing age. For BFL-9 aggregates, the expansion of microbar cured at 80 °C was 49% larger than that of microbar cured at 60 °C. When the curing temperature increased, the expansion of microbars became larger. Differences in the expansion of microbars cured at different temperature can be ascribed to the reaction rate of AAR. Increasing the curing temperature improved the rate of alkaline ion migration into the interior of dolomite and accelerated the AAR, thus promoting the expansion of modified microbars. Therefore, selecting 80 °C as the curing temperature can shorten the time for experiment.

Figure 4: The expansion of modified microbars prepared by BFL-1 (a) and BFL-9 (b) rocks cured at 20, 60 and 80 °C.

Effect of concentration of TMAH solution on the expansion of modified microbars

To further investigate the influence of the concentration of curing solution on the expansion of modified microbars, the samples prepared by BFL-1 and BFL-9 aggregates with 2.5-5 mm particle size at 80 °C were cured in 1 and 2 mol/L TMAH solutions, respectively. The results were given in Figure 5. From Figure 5(a), samples made by BFL-1 aggregates had small expansion at early curing age. After 14 days, the corresponding expansion became obvious. There was a shape increase in 56 d curing age. But the expansion of samples became stable after 112 days. It can be seen that

the differences in expansion between samples cured in 1 and 2 mol/L were slight. Compared to cured in 1 mol/L TMAH solution, samples cured in 2 mol/L had slight lower expansion before 70 d, but higher expansion appeared after 70 d. The expansion of BFL-1 samples cured in 1 and 2 mol/L TMAH solutions were 0.167% and 0.183% at 140 d. As shown in Figure 5(b), the expansion process of samples made by BFL-9 aggregates was just like to BFL-1 aggregates, except bigger expansion. The expansion of BFL-9 samples cured in 1 and 2 mol/L TMAH solutions were 0.186% and 0.194% at 140 d. No obvious differences between samples cured in 1 and 2 mol/L can be ascribed to the sufficient alkaline in these two concentrations of alkali solutions. 1 mol/L TMAH solution may provide enough alkaline in the process of alkali-dolomitic reaction. But at later curing age, the alkaline maybe not enough for the alkali-consuming ADK, which may accounts for samples cured in 2 mol/L TMAH solution with higher expansion than that cured in 1 mol/L TMAH solution. By taking expansion characteristics as well as economy into consideration, the particle size, curing temperature and concentration of TMAH solution were fixed at 2.5-5 mm, 80 °C and 1 mol/L, respectively, to investigate the microstructure of dolomitic aggregates cured in TMAH solution.

Figure 5: The expansion of modified microbars prepared by BFL-1 (a) and BFL-9 (b) rocks cured in 1 and 2 mol/L TMAH solutions.

Microstructure

In order to investigate effect of alkali-dolomitic reaction on the pore structures of dolomitic rocks, Figure 6 shows the pore size distribution of BFL-9 original aggregate grains and reacted aggregate grains cured in TMAH solution for 140 days. It can be seen that the pores in original aggregates with a pore diameter range of 0.2-1.1 μm , but the pores in reacted aggregates with a pore diameter range of 0.1-1.2 μm . Compared to the original aggregate grain in pore size distribution, the reacted aggregate grain had new-formed micro-pores with pore diameter ranging from 0.1-0.2 μm . As shown in Figure 6(b), the total porosities of original and reacted aggregates were 0.143% and 0.162%, respectively. Compared to the original aggregates in

porosity, the reacted aggregates exhibited a slight increase (13%) in total porosities at 140 d, which was similar to the results from P. Stukovnik [16]. This may be due to that the growth of brucite crystal as result of alkali-dolomite reaction in the restricted space dilated the pores.

Figure 6: Typical pore structure of BFL-9 aggregates: (a) derivative porosity curve (pore size distribution) and (b) cumulative porosity curve.

To characterize the reaction products of alkali-dolomite reaction, Figure 7 shows DSC-TG analysis on BFL-9 dolomitic grain cured in TMAH solution for 140 days. It can be seen that there was a slight peak at 380 °C, which represented the existence of brucite in reacted dolomitic aggregates. The slight peak is ascribed to the small amount of brucite. According to Tong [2], brucite crystals grow up in confined spaces, bringing about the expansion of dolomitic aggregates, which may accounts for the changes in total porosities. To further characterize brucite formed in the process of ADR, SEM-EDS analysis of dolomitic aggregate grains was carried out.

Figure 7: DTA-TG curves of BFL-9 dolomitic aggregate grain cured in TMAH solution for 140 d.

From Katayama [9], the alkali-dolomite reaction can occur with the reaction rims and along pre-existing cracks, and the characteristic myrmekitic texture can be visible. But to further investigate the reaction products of ADR, as shown in Figure 8, the reacted BFL-9 aggregate with dolomite enrichment area and cracks was selected to be observed by SEM-EDS analysis. From Figure 8(a), it can be seen that many rod-like crystals beside the micro-crack. Figure 8(b) shows EDS spectra determined in zone 1. Combining the two figures, it is indicated that rod-like brucite crystals formed in the process of ADR indeed exist. The distribution of brucite crystals can be explained by that the alkali-dolomite reaction can not only occur at the interface between the dolomite and the cement paste but also inside the dolomite. When the pH and temperature of the curing solution is high enough, the ADR can continue to occur. Due to TMAH solution utilized in this work to rule out the influence from microcrystalline quartz and exclude the expansion contribution of ASR, the expansion only origins from the alkali-dolomite reaction. So the expansion of the modified microbars cured in TMAH solutions can be ascribed to the existence and growth of brucite crystals in limited space although the dolomitization with a process of volume reduction. Based on the aforementioned discussion, it can be indicated that the alkali-dolomite reaction can occur in some special dolomitic aggregates cured in 1 mol/L TMAH solution at 80 °C and brings about expansion.

Figure 8: SEM-EDS analyses. (a) SEM image of BFL-9 aggregate with 2.5-5 mm grain cured in TMAH at 80 °C for 140 d. (b) EDS spectra determined in zone 1.

Conclusions

In this work, the modified microbars prepared by dolomitic aggregates and self-made cement were used to systematically investigate effects of the concentration of TMAH solution, curing temperature and aggregate grain size on the expansion of dolomitic aggregates. From the physical measurement and microstructural analysis, the following major conclusions can be drawn:

Changing the aggregate grain size for modified microbars and increasing the

curing temperature can considerably increase the rate of alkali-dolomite reaction and then accelerate the samples with bigger expansion. Compared with 0-1.25, 1.25-2.5 mm particle size, modified microbars prepared by 2.5-5 mm grains have bigger expansion. The expansion of microbars cured at 80 °C is 49% larger than that cured at 60 °C for 140 days, much larger than that cured at 20 °C. The expansion of microbars cured in 1 and 2 mol/L TMAH solution have no obvious differences, which probably due to adequate alkaline for ADR.

Based on the analysis on pore size distribution and total porosities of aggregate grains, the porosity increased 13% as result of ADR. According to the TG analysis assisted SEM-EDS analysis, it was revealed that rod-like brucite crystals formed in the process of ADR were found.

Due to TMAH solution utilized in this work to rule out the influence from microcrystalline quartz and exclude the expansion contribution of ASR, the expansion only origins from the alkali-dolomite reaction. The alkali-dolomite reaction can occur in some special dolomitic aggregates cured in 1 mol/L TMAH solution at 80 °C and brings about expansion.

Research ethics

This study uses no humans or human tissues.

Animal ethics

No animals are used in this work.

Permission to carry out fieldwork

This work has no needs for the removal of fossil specimens etc.

Data Accessibility

Our data are deposited at Dryad:
<https://datadryad.org/review?doi=doi:10.5061/dryad.https://doi.org/10.5061/dryad.c4n0b61>.

Authors' Contributions

B.Y. designed and conducted the experimental program, and drafted the manuscript. X.X.C. carried out the statistical analyses and collected field data. M.D. provided and designed the project. All authors contributed to the analysis and conclusion. And all authors gave final approval for publication.

Competing interests

We have no competing interests.

Funding

This work was supported by the National Key Research and Development Plan of China (2017YFB0309903-01) and the Priority Academic Program Development of Jiangsu Higher Education Institutions (PAPD).

Acknowledgments

The authors gratefully acknowledge the assistance from Mr Liu Peng and Mr Yu Qing from NJTECH, and the staffs from State Key Laboratory of Materials-Oriented Chemical Engineering.

References

- [1] Gill J.E. Petrology of Dolomitic Limestone, Kingston, Ontario, Canada. *Geo. Socie. Amer. Bull.* 1967, 759-778.
- [2] T.M. Tong Liang. Correlation between reaction and expansion of alkali-carbonate reaction, *Cement and Concrete Research* 25 (3) (1995) 470-476. (doi.org/10.1016/0008-8846(95)00034-A)
- [3] P.E. Grattan-Bellew, G. Chan, Comparison of the morphology of alkali-silica gel formed in limestones in concrete affected by the so-called alkali-carbonate reaction (ACR) and alkali-silica reaction (ASR), *Cement and Concrete Research* 47 (2013) 51-54. (doi:10.1016/j.cemconres.2013.01.013)
- [4] P.E. Grattan-Bellew, L.D. Mitchell, J. Margeson, D. Min, Is alkali-carbonate reaction just a variant of alkali-silica reaction ACR=ASR?, *Cement and Concrete Research* 40 (4) (2010) 556-562. (doi:10.1016/j.cemconres.2009.09.002)
- [5] M. Beyene, A. Snyder, R.J. Lee, M. Blaszkiewicz, Alkali Silica Reaction (ASR) as a root cause of distress in a concrete made from Alkali Carbonate Reaction (ACR) potentially susceptible aggregates, *Cement and Concrete Research* 51 (2013) 85-95(doi:10.1016/j.cemconres.2013.04.014)
- [6] Tang M.S., Han S.F., Liu Z. Mechanism of alkali-carbonate reaction. *Proceedings of the 7th International Conference on Alkali-Aggregate Reaction in Concrete*, Ottawa, Canada, August, 1986:275-279.
- [7] T. Katayama, M. Tagami, Y. Sarai, S. Izumi, T. Hira, Alkali-aggregate reaction under the influence of deicing salts in the Hokuriku district, Japan, *Materials Characterization* 53 (2-4) (2004) 105-122. (doi:10.1016/j.matchar.2004.07.003)
- [8] T. Katayama, How to identify carbonate rock reactions in concrete, *Materials Characterization* 53 (2-4) (2004) 85-104. (doi:10.1016/j.matchar.2004.07.002)
- [9] T. Katayama, The so-called alkali-carbonate reaction (ACR) — Its mineralogical and geochemical details, with special reference to ASR, *Cement and Concrete Research* 40 (4) (2010) 643-675.

(doi:10.1016/j.cemconres.2009.09.020)

[10] L. Z.K.L., Wang H.b., Chen Y. Exploration and protection of alkali-aggregate reaction, *Shanxi Architecture* 43(19) (2017) 117-119. (doi:10.13719/j.cnki.cn14-1279/tu.2017.19.063)

[11] ASTM C295: Standard Guide for Petrographic Examination of Aggregates for Concrete. Annual book of ASTM standards, 1992, 179-186.

[12] ASTM C586-11: Standard Test Method for Potential Alkali Reactivity of Carbonate Rocks as Concrete Aggregates.

[13] RILEM Recommendation AAR-5: Rapid preliminary screening test for carbonate aggregates.

[14] ASTM C1293-08: Standard test method for determination of length change of concrete due to alkali-silica reaction.

[15] T. Prinčič, P. Štukovnik, S. Pejovnik, G. De Schutter, V. Bokan Bosiljkov, Observations on dedolomitization of carbonate concrete aggregates, implications for ACR and expansion, *Cement and Concrete Research* 54 (2013) 151-160. (doi:10.1016/j.cemconres.2013.09.005)

[16] P. Štukovnik, T. Prinčič, R.S. Pejovnik, V. Bokan Bosiljkov, Alkali-carbonate reaction in concrete and its implications for a high rate of long-term compressive strength increase, *Construction and Building Materials* 50 (2014) 699-709. (doi:10.1016/j.conbuildmat.2013.10.007)

[17] Chen B, Deng M, Lan X, H. Behaviors of reactive silica and dolomite in tetramethyl ammonium hydroxide solutions. Proceedings of the 15th International Conference on Alkali-Aggregate Reactions in Concrete, St Paul, Brazil, July 3-7, 2016.

[18] P. Krivenko, R. Drochytka, A. Gelevera, E. Kavalerova, Mechanism of preventing the alkali-aggregate reaction in alkali activated cement concretes, *Cement and Concrete Composites* 45 (2014) 157-165. (doi:10.1016/j.cemconcomp.2013.10.003)

Appendix B

Responses to reviewer's comments:

Reviewer #1:

Reviewer's comments: *This is an interesting work that discusses TMAH solution was utilized as curing solution to investigate the expansion characteristics only caused by ADR. However, several aspects of the work must be improved before it can be accepted for publication. I have listed them in detail below:*

1. Details of TMAH solution are not presented. What are the similarities and differences between TMAH solution and NaOH solutions? such as PH? chemical formula?

Response: Thank you very much for your valuable comments and suggestions. The chemical formula of TMAH is $(\text{CH}_3)_4\text{NOH}$ with 91.15 relative molecular mass. It is very easy to absorb moisture and has a weak ammonia smell. TMAH solution is a colorless and transparent liquid with an ammonia smell as well. The degree of alkalinity is similar to that of NaOH. Compared to NaOH solution, TMAH solution belongs to organic alkali rather than common inorganic alkali. In addition, NaOH reacts with SiO_2 but TMAH does not react with SiO_2 . The details of TMAH solution have been added in the manuscript (please see p. 3 lines 23-27).

2. Most research about AAR(alkali-aggregate reaction) is based on NaOH solution, which cannot rule out the expansion originates from

ASR or ACR. But TMAH can exclude the expansion contribution of ASR. What is the effect mechanism? What are the differences between TMAH and NaOH resulted in the different expansion?

Response: Thank you for your question. TMAH was used to exclude the expansion contribution of ASR due to TMAH does not react with SiO₂. According to J.T.L. Thong ^[1], TMAH has been widely utilized in silicon anisotropic etching due to reliable etching of silicon in the fabrication of microelectronechanical systems. The reaction equation is shown in equation as follow:

From the reaction chemical formula, it can be seen that TMAH is used as reactant and SiO₂ is the corresponding product. It can be conclude that TMAH does not react with SiO₂ because reactants and products do not react. The differences between TMAH and NaOH resulted in expansion can be conclude two points. One is that the expansions of samples cured in TMAH solution are less than the expansions of the same samples cured in the NaOH solution due to excluding the expansion contribution of ASR. The other is that the expansions of samples cured in TMAH solution develop slower than that cured in NaOH at early age. The reasons for this phenomenon and research on the correlation between the expansion and the content of brucite are the key points in our future work. The effect mechanism and the differences in expansion have been added in this manuscript (please see p. 2 lines 38-42, p. 6 lines 25-28).

[1] J.T.L. Thong, W.K. Choi, TMAH etching of silicon and the interaction of etching parameters, Sensors and Actuators A 63 (1997) 243-249.

3. Table 1 the composition of cement clinker, CaCO_3 should be changed to CaO .

Response: Thank you for your comments. In fact, Table 1 shows the raw materials composition of cement clinker rather than the chemical composition of cement clinker. Analytic CaCO_3 , SiO_2 , Al_2O_3 and Fe_3O_4 were used as raw material to prepare cement clinker without K^+ and Na^+ in this work.

4. Page10Line31, which represented the existence of brucite in reacted dolomitic aggregates. Can you quantify the amounts of the brucite?

Response: Thanks for the reviewer's suggestion. According to your advice, we have quantified the amounts of the brucite by TG-DSC analysis. Fig. 7 shows the typical TG-DSC curves of the dolomitic aggregate grain cured in TMAH solution. There is a endothermic peak appearing at the DSC curves, namely from 350 °C to 400 °C, which is corresponding to the dehydration of $\text{Mg}(\text{OH})_2$. Therefore, the contents of $\text{Mg}(\text{OH})_2$ contained in the aggregate grain could be estimated according to the following equation:

$$\text{Mass}_{\text{Mg}(\text{OH})_2} = \frac{58 \times \text{Mass loss}(350^\circ\text{C} - 400^\circ\text{C})}{18}$$

where, the $\text{Mass}_{\text{Mg}(\text{OH})_2}$ refers to the contents of $\text{Mg}(\text{OH})_2$ formed in the process of alkali-carbonate reaction, and 58 and 18 represent the relative molecular mass of $\text{Mg}(\text{OH})_2$ and H_2O , respectively. On the basis of the abovementioned equation, the amounts of the brucite contained in BFL-9 dolomitic grain were 0.61%. The details of

quantifying the amounts of the brucit have been added in the manuscript (please see p. 9 lines 20-27, p. 10 line 1).

Reviewer #2:

Reviewer's comments:

1. *“This name is no common. Please indicate reference. I don't think so. Alkali silica reaction is the most common.” (Alkali-carbonate reaction (ACR) also called Alkali-dolomite reaction (ADR) is one of important kinds of alkali-aggregate reaction.)*

Response: Thank you very much for your valuable comments and suggestions. Alkali-dolomite reaction is no common in many references. “ACR is one of important kinds of alkali-aggregate reaction” is not suitable. We have revised our statement and ADR has been changed to ACR. The revised statement about ACR have been added in the manuscript (please see p. 1 line 29).

2. *“Myrmekite describes a vermicular, or wormy, intergrowth of quartz in plagioclase. I don't understand which is the characteristic myrmekitic texture in a dolstone?” (From Katayama, the alkali-dolomite reaction can occur within the reaction rims and along pre-existing cracks, and the characteristic myrmekitic texture can be visible.)*

Response: We failed to accurately express the views of Katayama. From Katayama,^[1, 2] according to polarizing microscopy, it was observed that dedolomitization produced a myrmekitic texture, composed of spotted brucite and calcite within the reaction rim, along with a carbonate halo of calcite in the surrounding cement paste. In addition, according to Štukovnik^[3], myrmekitic texture can be seen as

follow:

The characteristic myrmekitic texture is not the key point in this work. We have revised our statement in this manuscript (please see p. 10 lines 8-10).

[1] T. Katayama, How to identify carbonate rock reactions in concrete, *Materials Characterization* 53 (2-4) (2004) 85-104.

[2] T. Katayama, The so-called alkali-carbonate reaction (ACR) — Its mineralogical and geochemical details, with special reference to ASR, *Cement and Concrete Research* 40 (4) (2010) 643-675.

[3] P. Štukovnik, T. Prinčič, R.S. Pejovnik, V. Bokan Bosiljkov, Alkali-carbonate reaction in concrete and its implications for a high rate of long-term compressive strength increase, *Construction and Building Materials* 50 (2014) 699-709.

3. *“Brucite crystalization is a consequence of the dedolomitization process. If you have brucite would be associated with calcite and not with dolomite.”(The distribution of brucite crystals can be explained by that the alkali-dolomite reaction can not only occur at the interface between the dolomite and the cement paste but also inside the dolomite.)*

Response: Thanks for your comments. According to SEM-EDS

analysis, brucite can be seen in Figure 8(a). Brucite is the reaction product result from alkali-carbonate reaction. Dolomite reacts with alkali forming brucite and calcite, so they are coexisting. But in the process of dedolomitization, unreacted dolomite still exists in the aggregate grain because some dolomite have not yet fully reacted. We have revised this sentence (please see p. 10 lines 16-19).

4. *“Clarify the sentence” (So the expansion of the modified microbars cured in TMAH solutions can be ascribed to the existence and growth of brucite crystals in limited space although the dedolomitization with a process of volume reduction.)*

Response: Sorry for our inaccurate statement. In fact, the process of alkali-carbonate reaction can be conclude by the following two equations:

Firstly, dolomite contained in carbonate rock reacts with the alkali released from cement or admixture. Formula (1) is a reaction in which the solid phase volume is reduced, and the calculation shows that the solid phase volume is reduced by 4.4% ^[1]. Lastly, $\text{Ca}(\text{OH})_2$ formed in the process of cement hydration reacts with CO_3^{2-} , generating secondary calcite and OH^- . According to Katayama ^[2], the solid phase volume of formula (2) is increased by 10.2%. Therefore, our statement is inaccurate. According to Tong [3], brucite crystals grow up in confined spaces, bringing about the expansion of dolomitic aggregates. We have revised this sentence. Corresponding revision has been added

in this manuscript (please see p. 8 lines 22-24).

[1] X. Feng, N. Feng. Expansion mechanism of alkali-carbonate reaction [J].

Journal of the Chinese Ceramic Society, 2005, 3(7): 912-915.

[2] T Katayama. The so-called alkali-carbonate reaction (ACR)-Its mineralogical and geochemical details, with special reference to ASR [J]. Cement and Concrete

Research, 2010, 40(4): 643-675.

[3] L. Tong, Correlation between reaction and expansion of alkali-carbonate reaction,

Cement and Concrete Research 25 (3) (1995) 470-476.

In addition, some careless mistakes have been revised according to Reviewer's advice. The revisions were highlighted in red.